# Recommendations to Enhance Parental Involvement and Adolescent Participation in Physical Activity

**DOI:** 10.3390/ijerph19031333

**Published:** 2022-01-25

**Authors:** Colleen Cozett, Nicolette V. Roman

**Affiliations:** 1Department Sports Recreation and Exercise Science, Faculty of Community and Health Sciences, University of the Western Cape, Bellville, Cape Town 7535, South Africa; 2Centre for Interdisciplinary Studies of Children, Families and Society, Faculty of Community and Health Sciences, University of the Western Cape, Bellville, Cape Town 7535, South Africa; nroman@uwc.ac.za

**Keywords:** recommendations, adolescent, adolescence, physical activity, physical inactivity, parental involvement

## Abstract

Background: Adolescents are influenced by external factors which may impact their level of physical activity. Parents require specific strategies to become involved and to increase physical activity participation in adolescence. Objective: Thus, the current study aimed to design recommendations to increase physical activity participation and parental involvement. Methods: The current study forms part of a broader mixed-method study in which the results of the phases and stages of the pre-studies informed the current study. Thus, the current study uses an agreement workshop to develop recommendations with stakeholder and expert input in two rounds. Participants were invited to participate in the current study *n* = 100, and *n* = 65 participated in round one. Round two consisted of *n* = 20 experts invited to an agreement workshop, with *n* = 11 attending and making an input on the final recommendations. Therefore, experts and parents in the field of parenting, physical activity, and physical education, were invited to participate in the study rounds. After each round, the responses from the panellists were collated, interpreted, and developed into a framework for recommendations using thematic analysis. Themes were generated and refined using an agreement format. Results: After results from the stages and phases were consolidated and refined, six themes and 51 sub-themes were identified in a framework for recommendations. The framework was further refined using expert input and the final recommendations were derived using an agreement or agreement. Thus, with input from experts input through the agreement workshop, the findings were discussed, refined, and drafted into recommendations. Agreement and agreement were achieved on six broad recommendations and fifty-one sub-themes. The final recommendations were presented in the current study to increase parental involvement and physical activity in adolescents. Discussion: Recommendations and physical activity resources were developed and are presented as a form of support to parents and adolescents. The recommendations are intended as a source of unbiased information for parents to become more involved and for adolescents to increase physical activity participation.

## 1. Introduction

Adolescence is a transitional phase of growth and development between childhood and adulthood [1,2,3]. The World Health Organization defined an adolescent as any individual between ages 10 and 24 years, and adolescence as a period of life in which adolescents have health and developmental needs [4,5,6]. Adolescence can be a time of both disorientation and discovery, a transitional period that can raise questions of independence and identity as adolescents cultivate their sense of self, during which they make choices about physical activity (PA) behavior [7,8]. It is a time to develop healthy behaviors, knowledge, and skills that will be important later in life. Adolescents account for just over sixteen percent of the global population [1,2]. In reaching the study outcomes, it is important to understand the adolescent phase and the factors influencing physical activity (PA). The parents and immediate family steer the adolescent’s development. Furthermore, how adolescents overcome these challenges and how parents become involved in the PA experiences of adolescents are explored in the current study. All the linkages mentioned lead to the development strategies to increase PA participation and parent involvement in adolescence. Adolescent health has become a global health concern [3]. Understanding adolescent health is vital. Yet, limited empirical evidence on adolescent health presents a gap in the research, which allows for the development of strategies to improve health among adolescents [4]. Many adolescents in both the developed and developing countries do not meet the health-related recommendations of engaging in at least 60 min of moderate-to-vigorous daily [5]. Some adolescent health behaviors pose a threat to contracting non-communicable diseases [5]. The health-related behaviors that arise during adolescence can track into adulthood and have health implications [6,7]. Non-communicable diseases (NCDs) are also known as chronic diseases and tend to develop in the long term. NCDs are the result of a combination of genetic, physiological, environmental, and behavioral factors. The statistics on NCDs indicated that they account for more than a third of all deaths in South Africa [8]. Notably, there are increases in risks related to NCDs such as physical inactivity. The development of ill-health as a result of these risky health behaviors adopted throughout adolescence adds to the economic burden of government health aid.

Physical activity is defined as any bodily movement produced by skeletal muscles that require energy expenditure [9,10,11,12]. It includes activities undertaken while working, playing, carrying out household chores, travelling, and engaging in recreational pursuits. The term “physical activity” should not be confused with “exercise”. Exercise is a subcategory of PA that is planned, structured, repetitive, and aims to improve or maintain one or more components of physical fitness [13,14,15,16]. Globally, 23% of adults, and 81% of adolescents, do not do enough regular PA to meet the global requirements. In most countries, levels of inactivity are higher in girls compared with boys. Levels of inactivity increase with age. It has been shown that regular PA of 60 min during adolescence promotes health and wellbeing [17,18,19]. Therefore, ref. [20] recommend that adolescents do at least 60 min of moderate to vigorous-intensity PA daily. Physical inactivity and resultant lower energy expenditure contribute unequivocally to cardiovascular diseases, such as coronary artery disease and stroke, which are considered major causes of disability and mortality worldwide [21]. Moreover, there is strong empirical evidence that inactivity, obesity, and insulin resistance are significant risk factors for the development of Alzheimer’s Disease (AD) [22]. Whilst there is considerable evidence that aerobic training, such as running and dancing (and probably other types of aerobic training), may lower the risk of AD, notwithstanding, there is a paucity of evidence that dynamic resistance training, static resistance and general fitness lowers the risk of AD [21,22,23]. Despite the widely acknowledged benefits of PA, adolescents engage in far less PA than is recommended [15]. Physical inactivity is defined as doing very little or no PA at work, at home, for transport, or during leisure time. In [3], it was indicated that around 23% of adolescents were not active enough globally. In high-income countries, 26% of men and 35% of women were insufficiently physically active, as compared to 12% of men and 24% of women in low-income countries. Low or decreasing PA levels often correspond with a high or rising gross national product. The drop in PA is partly due to inaction during leisure time and sedentary behavior on the job and at home. Likewise, an increase in the use of “passive” modes of transportation also contributes to insufficient PA levels.

In Africa, the situation remains bleak: Only 8–35% of African adolescents engaged insufficient levels of PA for 60 min a day on at least 5 days per week [3,4]. Furthermore, PA has been estimated to be prevalent in 43–49% of South Africans adolescents. In South Africa, the PA of adolescents tends to decline with age and varies by gender, with boys reporting higher PA levels than girls [4]. While current health-related PA recommendations expect adolescents to be physically active in all domains of life, the majority of studies in Africa have focused mainly on adolescents’ overall PA levels. It has been reported from other African countries that less than 50% of adolescents between 13 and 15 years of age are physically active for at least 60 min a day on at least 3 days a week [3,4]. Similarly, in Nigeria, about 72% of school-going adolescents reported engaging in PA at least once a month, 59% engaged at moderate levels, and more than 50% engaged in low levels of PA (3). Insufficient PA is increasing amongst adolescents, with many studies in high-income countries reporting a consequent increase in overweight and obesity [10]. In the South African setting, there is a combined prevalence of overweight and obesity of 15–25% among participants between the ages of 10 and 20 years, a finding that is higher than anticipated. In low and middle-income countries, urbanization and the nutrition transition are largely responsible for the decrease in PA levels. Adolescents do not meet the current PA recommendations of 60 min of moderate to vigorous PA per day. The need for more PA surveillance data from Africa was highlighted [8].

The involvement of the parent shapes and influences health behaviors, including PA during adolescence [24,25,26,27]. Thus, the conceptual framework linked to the health promotion theory and the Ecological systems theory poses that PA is a lifelong endeavor [28,29,30]. A healthy lifestyle leads to health outcomes that can be beneficial until adolescents reach adulthood. Moreover, the role of the parent in the PA experience of adolescents must be acknowledged. Despite significant research on the broader aspects, parental involvement, parental support, influence, and engagement, the link with parental involvement remains understudied [31,32,33,34]. Several mechanisms through which parents can be involved in PA have been proposed [24,25,26,27]. More specifically, the parent’s involvement in the adolescent’s PA behavior is vital. According to [25], characteristics of parental involvement include [31,32,33,34] family cohesion, meaning how many people in the adolescent’s life understand them, how much the adolescent and family have fun together, and how much their family pays attention to them; [14] parental monitoring [15] parental directive behavior related to family rules and boundaries and parental encouragement. The family is the smallest functional unit that instils values in a community. Families refer to societal groups that are related by blood (kinship), adoption, foster care or the ties of marriage, including civil marriages, customary marriages, religious marriages, and domestic partnerships.” The communities within which families exist share characteristics such as social cohesion. Social cohesion is defined as “the result of building shared values and enabling people to have a sense of engagement in a common enterprise-facing shared challenge” [25,26,27]. Thus, parents can positively or negatively influence the PA behavior of the adolescent. Adolescents’ perceived sense of belonging was found to be beneficial for PA behavior. A recent systematic review highlighted the link between the parents, health behaviors, and parenting approaches. The review highlighted the gap in the literature focusing on the health behavior of adolescents as well as how parents are involved. Farooq [31] confirmed that parental involvement is vital in the PA health behavior in which adolescents engage. The influence of the parent to increase their child’s participation in PA was shown to be the strongest predictor of PA [32]. The family remains the nucleus in society and adolescents model the health behaviors of their parents [33]. Thus, parents can play an important role in their child’s experience of PA. Parental involvement in PA has a mutual benefit for both the adolescents and the parents. Parental involvement in sports helps with the enrichment of the parents’ sports knowledge and creates a link between the parent and adolescent. It opens a platform where parents understand the adolescent more. Parental involvement plays a role in strengthening parent-child interaction. The parent and the adolescent can talk about important issues related to the child’s sporting experience during training sessions and competitions. It also creates time for parent-to-child togetherness. Since parental involvement extends far from only providing the youth with resources, parents supporting adolescents at sports events carry out roles like transporting the child to the sporting facilities and acting as coaches. This offers a space where the child and the parents can spend more time together. Despite numerous advantages that emanate from parental involvement in youth sports and PA, there are reported cases where parental involvement has led to some negative ramifications. An example of this is that parental pressure may cause an adolescent to be less active. It takes a lot of resources to be a sports parent. Therefore, the sports parents are at risk of taking actions that may have negative outcomes.

Thus, the current study placed the adolescent in the context of the phase of adolescence. The issue of physical inactivity was highlighted and its impact on adolescents. Lastly, the role of the parents and parental involvement were described. Therefore, the current study aimed to develop recommendations for enhancing parent involvement and adolescent participation in PA, based on an agreement workshop design using a two-phased approach [35,36,37]. The two-phased approach was used to involve the participation of [38] a panel of experts, and [38] a panel of stakeholders within the field of physical activity, physical education, parenting, and community development in sport, to assist in reaching an agreement on the recommendations [39]. The current study forms part of a larger study in which the phases and stages of the study were used to develop recommendations. The larger study refers to a full thesis in fulfilment of a PhD in Philosophy. The phases and stages of the overarching study informed the development of recommendations through the processes of data collection and data analysis. The need for parental involvement was highlighted when guiding adolescents to increase physical activity (PA) [40,41,42]. The role of the parents stems from the foundation and parenting style employed by parents. According to [24,25,26,27], an adolescent’s well-being was measured through self-esteem (academic, social, emotional, family, and physical). The parent-child parenting style was measured through parental warmth and strictness, and the adolescents’ parents were classified into one of four groups (indulgent, authoritarian, authoritative, and neglectful). Remarkably, the greatest personal well-being was found for adolescents raised with higher parental warmth and lower parental strictness (i.e., indulgent), and the greatest social well-being was found for adolescents raised with higher parental warmth. Consistently, poorer personal well-being and social well-being were associated with less parental warmth (i.e., authoritarian and neglectful [31,32,33,34]. Parental involvement is defined as parents and other adults who work with or care for adolescents, contributing to developing the recommendations. These adults should be aware that, as children become adolescents, they typically reduce their activity levels. Thus, parents play an important role in providing age-appropriate opportunities for physical activity. In doing so, they help to lay a foundation for life-long, health-promoting physical activity. This influence becomes especially decisive in adolescence since there is evidence that it is a key stage in the adoption of healthy habits [1,2,3,4] and the strengthening of positive PA behavior that contributes to the improvement of adolescents’ current and future health [5,6,7]. These needs, which were identified and discussed in the finding’s chapters, were shared with the participants in the agreement workshop [41,42]. Through the agreement workshop, the findings were discussed, refined, and drafted into recommendations [35,36,37]. This framework informed the formulation of the recommendations presented in the current study. It is important to enlighten parents and professionals to understand parental involvement in increasing adolescent PA [29,30]. Thus, these recommendations were aimed at parents to increase involvement and adolescents to increase PA participation using an agreement workshop methodology.

## 2. Methods and Theoretical Framework

The current study was part of a broader study in which the stages and phases included participants as follows: Phase one, Stage one, review stage (Problem identification), Stage two: Quantitative stage (*n* = 1000 adolescents and *n* = 712 final sample), and Stage 3 Qualitative stage (*n* = 45 adolescents, and *n* = 35 final sample). Phase two, Stage one, Concept mapping and developing a framework for recommendations using an agreement workshop, Round one, *n* = 100 parents, stakeholders, and experts and *n* = 65 final sample (current study), Round two, *n* = 20 panel of experts and *n* = 11 attended the final agreement session (current study) and lastly the methodology was based on Stage two: Agreement workshop developing final recommendations. Figure 1 illustrates the broader study framework and the breakdown of the phases and stages to place the current study into context. Thus, the current study is the final manuscript linked to phase two of stage two (agreement workshop).

The current agreement workshop was based on the stages and phases depicted in Figure 1. Thus, an agreement workshop was used as the research methodology in the current study in which theory was built using the results and findings of the pre-studies as a foundation in the current study. Refs. [35,36,37,38,39], stated in recent findings on workshops that it is a research methodology that is reliable in producing valid data as it aims to achieve participants’ expectancies and to provide reliable information. Therefore, this research study was concerned with including the parents and experts by sharing knowledge to define and understand a problem to find solutions [26,27].

## 3. Design

An agreement workshop design was conducted to develop recommendations to enhance parent involvement and PA participation in adolescence. The objective of the agreement workshop was to present the key findings of the pre-study with input from a panel of experts and stakeholders and work towards reaching an agreement [36,37]. The agreement workshop was best suited for agreement building and was based on the assumption that group judgments are more convincing than individual judgments [38,39,40]. During each round, once group agreement was reached, the process was stopped. This study received ethical approval from the Research Ethics Committee at the University of the Western Cape (ethical clearance number HSS: 17/10/16). Thus, the pre-study findings were consolidated in a concept map, and themes and sub-themes were identified using the framework for recommendations to inform the development of recommendations to increase parent involvement and PA in adolescence.

### Inclusion and Exclusion Criteria

Participants in the current study were as follows: participants who gave consent to participate in the study by completing the participant confidentiality sheet. Furthermore, participants who gave consent and attended the actual session. Participants who were experts in the field of physical activity, parenting, research family studies, and physical education were purposively selected to be in the study. The sessions were held online on Zoom due to the lock-down procedure that existed in the South African context. Participants who did not give consent and who did not attend the online sessions were excluded.

## 4. Round One: Participant Selection

To ensure a broad perspective on the themes, experts from the schools, communities, and the University of the Western Cape working in the field of physical activity, adolescent health, education, physical education, research, and Sport Science as well as in community development who had relevant knowledge and expertise at the research, clinical and policy level, were invited via email to participate in round one [38]. On the day of the workshop, eleven of the panel of experts attended the workshop online due to the risk protocol in place due to COVID-19. The agreement framework below spells out the participant selection in each round of the study Figure 2.

## 5. Data Collection

A panel of experts (experts and academics) participated in round one (*n* = 65), to identify the unclear or ambiguous recommendations as indicated in Figure 2 of the research study. The goals of round one were to share with the panel of experts: (1) the aim and objectives of the current study; (2) the outcomes of each stage in Phase 1 of the research process, which resulted in the 36 recommendations; and based on these, (3) identify themes and sub-themes resonating with the findings. The panel of experts was asked to reach an agreement on each recommended guideline. In round two, *n* = 20 experts were invited to participate to reach an agreement on the final recommendations. Only *n* = 11 attended the actual session as indicated in Figure 2. Data collection included participants in the field of PA, physical education, and parenting.

### 5.1. Recommendations from the Ecological Systems Theory and Health Promotion Approaches

The current study was developed using an agreement workshop and was grounded in the Health Promotion theory and the Ecological systems theory. It is vital to view the adolescent in the context of the environment and influencing factors that may impact health and wellbeing [28,29,30]. From a Health promotion theorist’s perspective, recommendations were developed. The various phases and stages in this study informed the recommendations developed. Adolescents participating in health-enhancing PA produce health benefits [9,10,11,12]. In this document, the term “physical activity” generally refers to health-enhancing physical activity [9]. Brisk walking, jumping rope, dancing, lifting weights, climbing, cycling, and doing yoga are all examples of physical activity [12,13]. Lifestyle activities that encourage the increase of baseline PA to increase adolescent PA are essential for health [9,10,11]. Short bouts of activity are beneficial and can accumulate to the recommended number of minutes adolescents must participate in PA [1,2,3,4]. In that way, adolescents can accumulate the recommended daily minutes [9]. Therefore, the availability of infrastructure to support short bouts or episodes of activity is important [9,10,11,12,13,14]. For example, adolescents should have the option of using sidewalks and paths to walk between buildings at a worksite, rather than having to drive [1,2,3,4]. Therefore, adolescents must be allowed to make healthy lifestyle choices by selecting alternative options [8]. Physical activity is considered an important determinant of health, quality of life, and well-being. World Health Organization [6] defines physical activity as any bodily movement produced by skeletal muscles that require energy expenditure. Physical activity can also be undertaken as a form of movement-transport or house cleaning duties and through play [14,15,16]. All forms of physical activity can provide health benefits if undertaken regularly and of sufficient duration and intensity [15,16,17,18]. Physical activity is further identified as an essential component of health and inactivity in adolescence may track into adulthood [9]. Health behavior decisions are when an adolescent had the option of taking the stairs instead of using an elevator [9]. Moreover, PA that is fun, social and leads to enjoyment of the outdoors, improve their PA participation [10,11,12]. The proposed recommendations encourage PA for any meaningful reasons and the health promotion theory lays the foundation that PA participation should be lifelong. Therefore, the recommendations were developed with a lifespan approach and provided recommendations for adolescents and parents to encourage long-term participation. Physical Activity recommendations must be straightforward and clear while remaining consistent with complex scientific information.

### 5.2. The Ecological Systems Theory in Practice

The conceptual framework was based secondly on the Ecological Systems Theory which provided the theoretical foundation on which the recommendations were formulated [28,29,30]. Thus, the link with the external environmental factors is vital to understanding in the case of adolescents. The Ecological Systems Theory was deemed the appropriate theory as it focused holistically on the adolescents, the parents, and the environment. Therefore, adolescents and the recommendations developed are put into practice and action. The Ecological systems theory is based on a model that looks at different levels and how it influences the adolescent. The levels in practice are the individual level (Microsystem), Relationship level (Meso-system), Community level (Exo-system), and the societal level (Macro-system). The Chrono-system is linked to the specific phase of adolescence. The adolescent’s developmental phase plays a role in the individual’s and in the context of the current study the adolescents were aged 15 years. Adolescents and parents are influenced by various factors. The factors that influence the unit enable PA or are PA barriers [24,25,26,27]. Thus, the recommendations in practice will provide the parents and adolescents with a resource to overcome the barriers and environmental challenges that hamper PA. In South Africa, the family is viewed as a social system because its members are interdependent and any change in the behavior of one member will affect the behavior of others [26]. The recommendations designed in the current study intend to provide support to parents to become more involved and adolescents to increase participation. There has been an acknowledgment of the importance of strengthening and building the capacity of parents and caregivers to support adolescents [31,32,33,34]. In terms of the bigger picture, adolescents model the behavior of adults in their lives [8]. An agreement workshop design was used to develop recommendations for reaching an agreement on the proposed recommendations. The objective of the agreement workshop was to present the key findings of Phase 1 and engage with the panel of experts and stakeholders and work towards reaching an agreement. Building agreement was the best method to take decisions on the recommendations for inclusion and was based on group judgments which are more convincing than individual judgments [37,38,39]. During each round, once group agreement was reached, the process was stopped. Furthermore, Figure 3 below illustrates the adolescent and how the various systems play a role such as the Macro, Micro, and Meso systems.

### 5.3. Participant Selection and Data Collection Processes

Figure 1 illustrates the participant selection and data collection in each round. The participant selection process took place in each phase and round of the current study using purposive sampling methods. Figure 1: The agreement framework is found in the addendum for the current study. The participants in the final selection included 20 experts and parents and the researchers from the University of the Western Cape working in the field of physical activity, adolescent health, education, physical education, research, and Sport Science. The panel had a wealth of knowledge in the field of community development and expertise in physical activity at the level of research, clinical, and policy level. The stakeholders were invited via email to participate in the round [41,42]. On the day of the workshop, eleven of the panel of experts attended the workshop online due to the risk protocol in place due to COVID-19 and *n* = 11 participated.

## 6. Results

The findings of the rounds, phases and stages focussed on six themes which were identified: key recommended recommendations for adolescents; recommendations for parents; information and resource support; recommendations related to increasing parental involvement; recommendations related to safety and PA environment; sustained PA to take action. Based on these key findings of the current study, the developed recommendations intended to increase parental involvement and PA participation in adolescents. An agreement was reached. Based on the 12 recommendations and 66 sub-themes recommended in Phase 1, the three themes and their corresponding sub-themes that were identified in Phase 1 were agreed upon by the panel of experts. However, the panel strongly argued for the inclusion of a list of recommendations specifying the actual activities for increasing physical activity and to provide parents and adolescents with specific support due to clarity needed with the terms and concepts of physical activity. Thus, the proposed 12 recommendations were consolidated into six themes and 51 sub-themes are presented in summary Table 2, meaning 14 items were removed and reasons are indicated in the table (items removed: 5, 6, 13, 14, 15, 34, 42, 44, 46, 47, 51, 53, 52, 62). The six themes included due to recommendations made by panellists were namely:1Six themes were identified after stakeholder and expert input in the final round:2Key recommended Recommendations for adolescents3Recommendations for parents4Information and resource support5Recommendations related to increasing parental involvement6Recommendations related to safety and PA environment7Sustained PA to take-action

Table 1 illustrates the themes and sub-themes in the framework for recommendations with all the items included. The 14 items have already been removed (5, 6, 13, 14, 15, 34, 42, 44, 46, 47, 51, 53, 52, 62) after stakeholder and expert input was made.

### Guideline Development

Responses by participants led to the 66 recommendations made in Phase 1 of the study, and responses to the set of questions corresponded with these recommendations from a sample of *n* = 65. Further suggestions were received from the 11-member panel of experts in the second round. Due to similar and overlapping themes, the panel of experts agreed to the merging of several of the recommendations proposed in round one, which resulted in 12 themes consolidated and 14 sub-themes were removed. The 12 main themes were consolidated into six themes and 51 sub-themes were included. Furthermore, based on evidence from round one, the agreement was reached to include a parental resource Appendix A under the overarching theme of resource support. In essence, there was unanimous agreement by all the experts concerning these recommendations. Additional comments from the panel of experts included a request for clarification on the term: “PA definitions in the resource table included.” The researcher combined the feedback on the recommendations made and included it in the report. The second round elicited a further two recommendations from the 11-member panel of stakeholders. The addition of the two recommendations was based on the evidence of the research undertaken in round one. Participants felt that parents should be made aware of the benefits of PA and definitions of PA. Table 1 illustrates the progress and recommendations made through the 2-round Agreement workshop process leading to the final framework for recommendations. The comments from participants and confirmations of recommendations for inclusion are included in Table 1. The recommendations below were designed for adolescents. Thus, recommendations were developed for parent’s involvement and adolescents according to (1) Key recommended Recommendations for adolescents, (2) Recommendations for parents, (3) Information and resource support, (4) Recommendations related to increasing parental involvement, (5) Recommendations related to safety and PA environment, and (6) Sustained PA to take action. An agreement rating below 70% was excluded.

## 7. Key Recommendations Are Presented

### 7.1. Recommendations for Adolescents

What parents need to know about guiding adolescents is that understanding physical activity concepts and definitions and being involved by supporting adolescents can be a daunting task. It may leave parents feeling uncertain about what the correct information is that applies to increasing physical activity in adolescents. Parents commonly ask questions and make statements such as: What can I do to help?; I need the information to help my child/adolescent. Consequently, they may have many unanswered questions. Questions asked by parents frequently include: “How long should my child be active?”; “What do you mean by recommended minutes of activity?”; “What activities are appropriate for an adolescent?” What parents need to know is that adolescents need to accumulate 60 min of activity of 20 min bout/sessions of activity to accumulate the recommended daily minutes. Regular participation in physical activity (PA) in adolescence can help reduce the risk of several chronic diseases (e.g., cardiovascular diseases, diabetes, certain cancers, hypertension, osteoporosis) and premature death in youth [31]. It can also promote healthy physical (e.g., build muscle, improve flexibility, maintain a healthy weight), psychological (e.g., reduce symptoms of stress, anxiety, and depression, enhance self-esteem), and social development (e.g., foster supportive relationships, reinforce a sense of belonging) in adolescents [24,25,26,27]. Physically active adolescents have reduced symptoms of anxiety and depression [12,13,14]. Moreover, physical activity is associated with improved mental health, and improved quality of life [17,18,19]. Physical activity recommendations provide parents with PA guidance about time, duration, intensity, and types of activities. The involvement of parents to create opportunities for PA and to support PA recommendations to increase PA is pivotal Centers for Disease Control and Prevention [21]. Therefore, the following six recommendations and sub-recommendations are stipulated for adolescents and parents below.

### 7.2. Key Recommendations for Adolescents

Recommended physical activity (PA) recommendations for adolescents include moderate- and vigorous-intensity physical activity for periods that add up to 60 min (1 h) or more each day. The concept of “informed choice” is fundamental, as parents need comprehensive, meaningful, and evidence-based information to make the appropriate choices when it comes to their child.Recommended physical activity (PA) recommendations for parents include moderate- and vigorous-intensity physical activity for periods that add up to 150–300 min for 3–5 days per week.Adolescents should do: 60 min of Physical Activity (PA) a Parents should participate for 150–300 min for 3–5 days per week.

### 7.3. Recommendations for Parents

Parental involvement in the adolescent’s early years of development makes a positive difference and enhances family and communication interaction [25].

Parents should help adolescents to set realistic goals. The goals set should assist adolescents to reach short-term, medium-term, and long-term PA goals. Goals should be realistic in that it is measurable, reachable, and attainable. Help adolescents to set realistic PA goals.Parents are instrumental in ensuring that free play remains a popular option to select as PA. Parents can create safe opportunities and supervise activities to ensure that adolescents can play freely. In this way, parents and adolescents devise plans together and opportunities for free and social PA settings.It is important for parents to ensure that PA and social activities created for parents are based on a variety of options. Adolescents need fun and variety to keep them invested in being active. Changing it up and surprising adolescents doing a variety of activities can help adolescents to have a selection of safe options to choose from to be active.

### 7.4. What Are the Activities to Do with My Adolescence?

The answer to the question above is in Table 2, which was developed specifically for parents to guide adolescents. The table developed by stakeholders, parents, and researchers provides specific recommendations about the following: type of activities, duration of time to participate, the intensity of how hard the adolescent must work. Parents can be involved in the PA lives of adolescents in the following ways:

### 7.5. Parental Involvement Recommendations

Be involved with planning for PA and preparation before events. Meaning, if the adolescent is participating in a school or community sport or PA event, parents can assist and get involved. If the parents know the school schedule, dates, and times on a notice board or fridge, iron the uniform, pack the sports bag, a reminder of practise or game times, prepare and buy snacks.Be actively involved in school sport.Parents can be actively involved by checking in with the coach.Parents can be involved by responding to school letters.Parents can be involved by volunteering to assist at school sports events if possible.Parents become spectators. Adolescents feel encouraged and supported if their parents watch their games. Go and watch them participate (spectator parents).

### 7.6. Parental Involvement (Support Recommendations)

Adolescents feel supported if parents are involved by participating in PA with adolescents for fun. Participating in PA with adolescents can become a bonding experience for parents and adolescents. It allows for adolescents to test their boundaries in a fun way and parents can oversee but take a lesser role in a game and let adolescents lead.Parents can support adolescents by buying their sports equipment, fees for transport to events, taking them to events, financial support, support with overcoming barriers to participation. Support adolescents by showing an interest in their PA.Parents can become more involved and show support by respecting adolescents’ choices for PA. Therefore, parents could guide adolescents, but, at times allow adolescents to select their PA and make their own choice too. It will teach adolescents PA autonomy.Parents should show support by ensuring that adolescents wear the correct personal protective gear or masks is something worn during COVID 19. Other types of protection in sport refer to a specific body part: (helmets, eyewear, goggles, shin-guards, elbow and knee pads, and mouth-guards masks).

### 7.7. Parental Involvement Recommendations (Directing Behavior)

Adolescents are in a phase of life where they tend to test boundaries. Therefore, it is important that parents set realistic boundaries for PA participation and in terms of rules. Rules and boundaries need to extend inside and outside the family home. The reason for this is that adolescents face a multitude of factors that influence them personally, socially, and environmentally. Therefore, rules and boundaries need to be set and parents and adolescents can collaborate and take ownership for actions taken.Parents need to establish open communication channels with adolescents in a collaborative spirit to develop ground rules, curfews, boundaries, PA time, limit screen time/sedentary time. If the decisions and consequences are agreed upon by all parties the chances that it will be sustained are more likely. Parents need to be consistent when applying decisions.Adolescents are savvy and parents are quick to note that at the times they find out their adolescent has skills unknown to them. Parents are advised to listen to their adolescents as adolescents are aware that they are adapting to an ever-changing environment under challenging circumstances. Therefore, let us give adolescents some credit; trust them to navigate the challenges by learning to take appropriate PA decisions.

### 7.8. Parental Involvement (Encouragement)

Adolescents model the physical activity behavior of parents. Being active parents allows the adolescent to model positive behavior and it is therefore encouraged. Good role models such as parents, caregivers, and teachers encourage an active lifestyle for children.Parents are a source of encouragement to adolescents and do it through praise, reward, and encouragement of positive PA behavior.

### 7.9. Parental Involvement (Awareness of Physical Activity Benefits)

Adolescents who are physically active experience the following benefits:Adolescents experience aerobic (heart) and muscle-strengthening physical activity which is beneficial long-term.Physical activity leads to health benefits for people with chronic and health conditions such as high blood pressure and diabetes and a reduced risk of cancer.Adolescents who participate get to feel good because of the release of endorphins.Improved self-concept, self-image, and reduced anxiety and depression risk. Improved sleep and quality of life.Physical activity creates an opportunity for positive peer interaction and social acceptance. Therefore, it helps adolescents feel like they fit in socially.

### 7.10. Personal and Community Safety

Teach adolescents to be aware of their surroundings and to be alert and avoid risky situations. Establish a buddy system for PA.Join a walking bus to and from school. Parents assist adolescents with sensible choices (when, where, how to be active).Choose places that are well lit. Following rules and safety rules is the best way to reduce activity-related injuries. Physical separation from motor vehicles and awareness of surroundings.

### 7.11. Sustained Physical Activity Recommendations (Taking-Action)

Provide time for both structured (formal) and unstructured (fun) physical activity.Physical activity such as online activities, active digital games, yoga, games, programs would interest adolescents.The use of technology and digital tools to use during sessions is preferred by adolescents.Parents and adolescents should remember to start any new activity gradually and be consistent. It takes approximately six weeks to truly say that a lifestyle change has happened.Physically active lifestyle changes must be made from a young age and stick with it.Everyone has a role to play: schools; communities; faith groups; businesses; civic organizations; parent-teacher associations; health groups; public safety agencies; policymakers.

### 7.12. Resource Support and Recommendations for Parents

It is important that parents are provided with information about how to support adolescents in enhancing PA participation. This will in turn assist parents to be more confident and aware of the strategies to use to increase PA by getting more involved in the needs of the adolescents.Information in the form of recommendations will assist parents to cope and enable them to make more informed decisions and choices, and thereby become actively involved in all aspects in setting objects for adolescents, preparing for participation, planning, being aware of the benefits of PA and safety precautions and how to ensure PA safety. Parents should be proactive and ask questions, as information in the form of recommendations can assist parents to become self-reliant, leading to parent confidence and exercising of own judgment in line with informed (parenting) choices.Parents must be informed and become the expert in the home to provide support, encourage, guide, and set realistic boundaries for PA. They should not readily accept or rely on professionals to make decisions on their behalf. Instead, the sourcing of information such as the current recommendations provides parents with an opportunity to have the information.In becoming the experts of their children’s PA future, parents should elicit professional support that is empathetic and unbiased to adolescents. Often, parents become strongly influenced by the information they receive from professionals and tend to follow such information relentlessly. In the case of the adolescent, we suggest that PA should be fun and a variety of strategies are possible to achieve success.The current Table 2 clarifies basic activities and provides an easy-to-follow template of possible activities for parents to guide adolescents.

The resource Table 2, p. 14 provides information regarding:frequency of activitiesduration of activitiesintensity of activity

### 7.13. Support Recommendations to Parents

Support to parents should be objective and unbiased information must be provided.The current recommendations could be provided in an article format, a poster format, a flyer format, electronic format on WhatsApp, by sharing information in live videos to reach parents or newsletters printed and electronic versions, visual and electronic formats.Regardless of the modality of communication chosen, support the communication option chosen by parents.Adopt an open and flexible attitude that reflects a non-judgmental approach to parents’ decisions on the communication options for their adolescents.Make every attempt to spread awareness by finding innovative ways to reach parents and share information, regardless of their socio-economic status, income, or geographic location.

## 8. Discussion

The purpose of the current study based on an agreement workshop was to develop recommendations for increasing parental involvement and increase PA participation in adolescence. This was successfully done through a workshop methodological approach using principles of action research in rounds, stages, and phases. Action research offers an alternative to knowledge development. It offers marginalized groups the opportunity to improve their situation [35,36,37,38]. Through the participation and collaboration of 11-members in a panel of experts and a 65 member panel of stakeholders, the emerged framework can be viewed as the first port of initial support for parents and stakeholders. In meeting the aim of the research study, the agreement was reached that the following agreed-upon recommendations: (1) Key recommended Recommendations for adolescents, (2) recommendations for parents, (3) information and resource support, (4) Recommendations related to increasing parental involvement, (5) Recommendations related to safety and PA environment, (6) Sustained PA to take-action options be integrated into a framework for recommendations for parents parenting adolescents to increase PA. The final recommendations developed through a process of the agreement provide a parent-centred and adolescent-centred approach to guideline development. Input into the development of recommendations was made by adolescents in phase one and parents in phase 1. Some of our findings have specific bearings on [24,25,26,27] research, such as how it relates to social and emotional support, (3) informed choices and (5) collaboration between parents and professionals. Firstly, assent was reached on resource support to parents. The motivation for this inclusion as agreed upon by all panellists was that parents should be provided with guidance, information and that offer parents support with regards to increasing PA and the type of activities to be included in the resource Table 2. Furthermore, panel 1 also indicated that this guideline has the potential to influence policy outcomes. These recommendations collaborate with the findings of [31,32,33] who put forward 12 best practice recommendations for infusing parent-professional partnership in the best interest of the adolescent. Secondly, the agreement was reached on Theme 1 and Theme 2 on recommendations for adolescents and recommendations for parents. All the panellists agreed that adolescents and parents should be provided with recommendations specific to their age and preferences. The aim of providing recommendations for parents is to encourage parents to be involved and to participate in PA with adolescents [25,26]. Our findings show that parents should receive unbiased support, which includes, guidance and recommendations that can increase PA in adolescents. These recommended recommendations have a strong correlation between social support for parents and increasing PA in adolescence [1,2,3,4]. Further recommendations are to include support strategies targeting adolescents to increase PA. Previous studies have found that parents’ health behavior influences adolescents’ behavior [5,6,7,8]. The inclusion of this recommendation in the recommendations will go a long way in contributing to the development and increasing PA. In addition, panel two further recognized the need in the recommendations for parents to understand their parenting support to adolescents in theme 4. Several studies have suggested that raising a child requires parents to adapt their parenting styles and skills, which would affect the quality of the parent-child relationship and enhance parental involvement in PA. Our framework for recommendations, therefore, has the potential to introduce parents to different parenting styles and to inform them of their role in parenting their adolescents. Thirdly, panels 1 and 2 reached an agreement to include a resource list in the framework for recommendations. This list encompasses accurate information and is well-balanced [18,19]. Such information allows parents to make informed choices and enables them to play an active role in their child’s development. In addition, a resource list provided a detailed explanation of the activities needed to increase PA. A resource list allows for collaboration and partnerships between professionals and parents [40,41,42]. The panel of experts felt that parents should be seen as partners and not mere receivers of information. In the fourth instance, the recommendations were that all safety-related recommendations be consolidated into one section. It became clear that safety was a priority to parents, and experts. A clear distinction was made between personal safety, environmental safety, and safety in the community. Our findings suggest that parents and adolescents must receive guidance and support about safety in the community. Hence, the agreement among the panel of participants highlighted the view that parents should be provided with objective information on a full range of safety options and guidance in the recommendations [41,42]. It is envisaged that this study will provide parents and professionals in the field with clear recommendations to be integrated into program delivery and policy development. Furthermore, the study can be viewed as a contribution to the field of physical activity and increasing PA in adolescents. In this study, attempts were made to select expert and stakeholder panellists who represented disciplines and constituencies relevant to PA, physical education, Sport Science, and parenting adolescents to increase PA. Parents who participated in the current study (round two) also participated in Phase 1 of the study. Likewise, the findings of the current agreement workshop highlighted experts’ and stakeholders’ participation and collaboration to reach an agreement on a set of recommendations to enhance PA participation and increase activity. The participation of experts and stakeholders in the design of the research study underscores the rigor undertaken in reaching an agreement in the development of a framework for recommendations. Having reached an agreement on a framework for recommendations, these findings may stimulate practical implementation, thus leading to further research opportunities. These applications include the need for providing training to parents, or the facilitation of parent to adolescent support groups, specific programs engaging fathers and mothers, connecting parents with adolescents, or programs within the domain of adolescent PA. With an understanding of the diverse needs of adolescents and parents in mind, these are but a few insightful recommendations made to facilitate and increase PA.

## 9. Conclusions

The research study was successfully implemented by employing an agreement workshop to develop recommendations. These recommendations have been designed to enhance parent involvement and increase adolescent participation in PA. Their purpose is to foster parents’ and adolescents’ understanding of the risks of an inactive lifestyle and how to find ways and strategies to get involved to support adolescents to increase PA. The 6 recommendations highlighted support for parents and adolescents, namely: (1) Key recommended Recommendations for adolescents, (2) Recommendations for parents, (3) Information and resource support, (4) Recommendations related to increasing parental involvement, (5) Recommendations related to safety and PA environment, (6) Sustained PA to take action. It is hoped that these six recommendations and specific focus areas will provide parents and adolescents with support strategies related to increasing PA and enhancing parental involvement. Furthermore, it includes parents and practitioners working collaboratively in the best interest of the adolescent, which promotes a better understanding of the diverse needs of parents and adolescents to increase PA. It allows professionals to develop an awareness of increasing PA, parental involvement and to build upon strengths to meet adolescents’ needs.

## 10. Recommendations for Practitioners

It is recommended that:Sports science and practice build on the insights generated herein and incorporate these recommendations in their practices, focusing on specific areas relevant to parent involvement to increase PA participation. For instance: (1) assisting parents to adapt their parenting approaches to support adolescents increasing PA participation; (2) emphasizing and focusing on the barriers to PA and the challenges adolescents are confronted with when participating in PA, and (3) creating awareness of the benefits of PA.Professionals (researchers, educators, sports scientists, parents and practitioners, and community workers) working in this context obtain a deeper understanding of the unique experiences and needs of adolescents and thereby provide support to adolescents through guideline development.Parents obtain the prerequisite skills to provide support to adolescents through resources provided in recommendations.Continued training and curriculum development take place in the field of PA, with specialized knowledge and skills in parent involvement and increasing PA in adolescence.Adolescent–parent-centred support should be linked to macro systemic, meso, and micro support, such as linked to the adolescent.The Department of Education (WCED) uses the insights generated herein as a framework or foundation to design and develop relevant and suitable programs to meet the needs of adolescents and parents to increase PA.

## 11. Recommendations for Future Research

It is further recommended that future research be conducted on:Overcoming barriers to PA and understanding the needs of adolescents to increase PA.Parenting programs targeting parents and adolescents.The role of the mother in PA support offers valuable and unique perspectives and insights.The role of the father in PA support offers valuable and unique perspectives and insights.The experiences of adolescents with inactive parents. Insight into their lives and growing up in a household with active or inactive parents is an important resource for parents and adolescents.Comparative studies comparing the experiences of adolescents in different contexts maybe provide invaluable research and insights.The topic in different geographical areas in South Africa, to identify trends as well as compare parents’ and adolescents’ challenges, experiences, perceptions, and needs.The topic through collaborative research at the university level, among organizations of the PA, not only in South Africa but across the African continent.Using technology as a strategy to increase PA.

## 12. Study Limitations

The following limitations were encountered in this study:COVID-19 became a limitation that had to be overcome. The lockdown period occurred amid the data collection and final workshop rounds in the current study. Therefore, major rearrangements had to be made to ensure social distancing was adhered to and protocol followed. Therefore, the workshop was held on Zoom and recorded for transcription.The study was conducted in Cape Town, one region of South Africa. The findings may therefore not be generalizable to other regions of the country.

## Figures and Tables

**Figure 1 ijerph-19-01333-f001:**
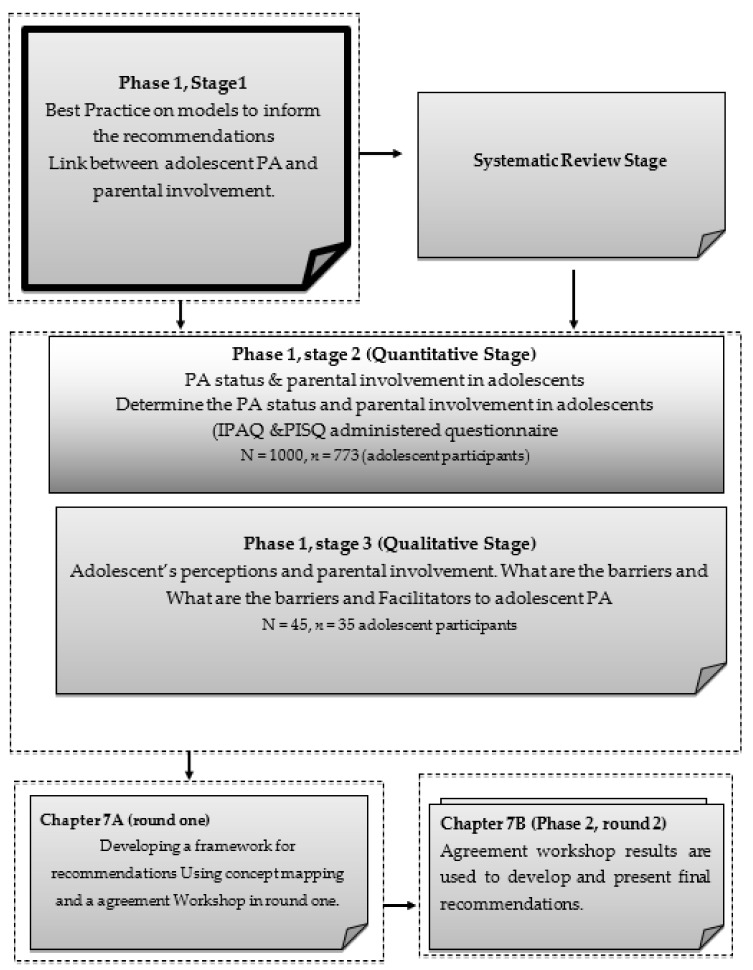
The pre-study plan.

**Figure 2 ijerph-19-01333-f002:**
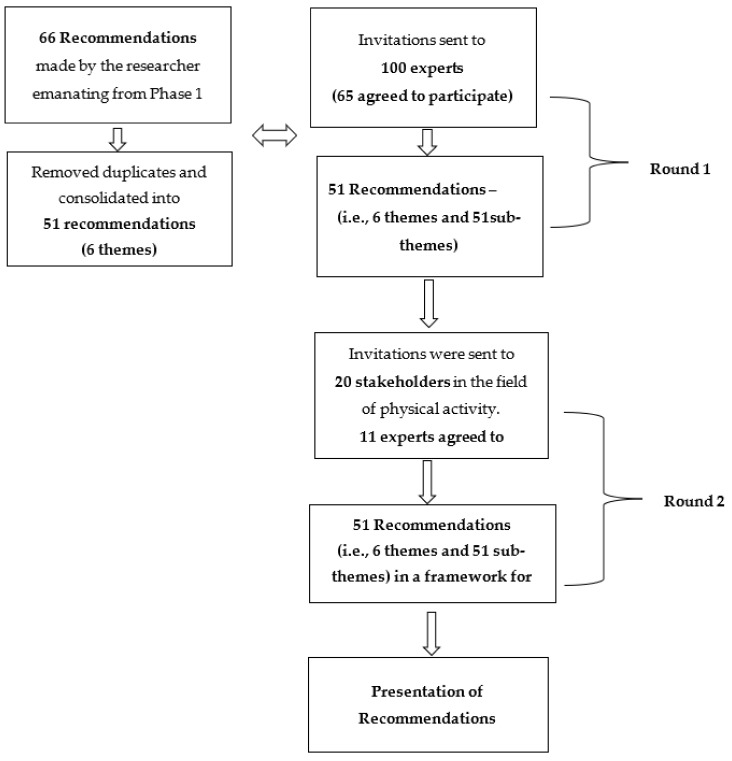
Agreement framework.

**Figure 3 ijerph-19-01333-f003:**
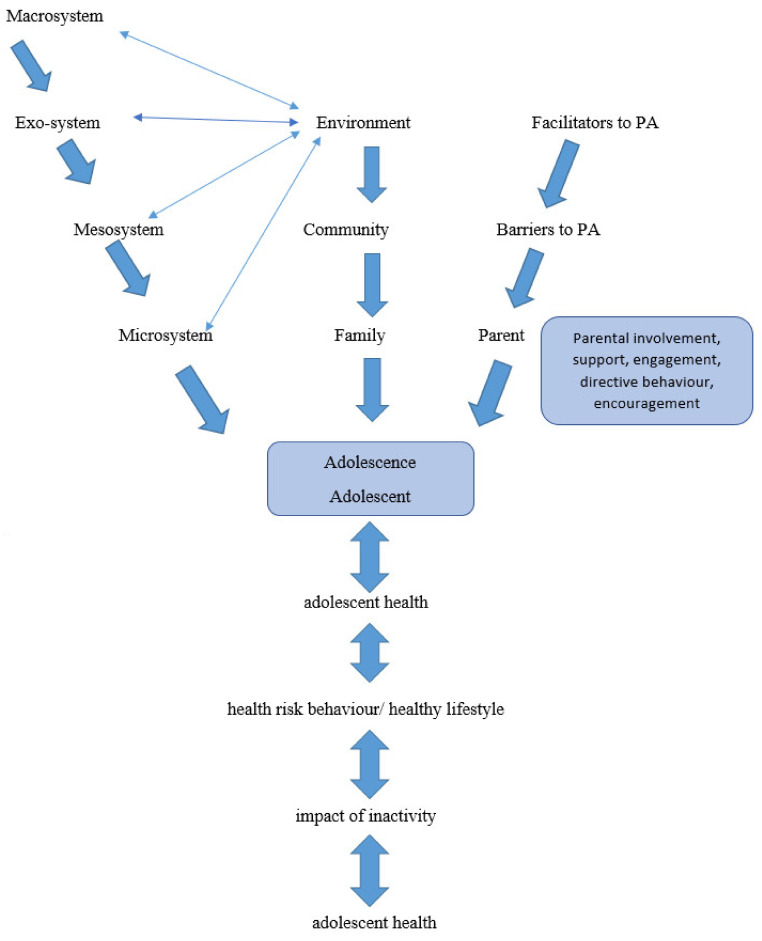
Environmental factors impacting adolescents.

**Table 1 ijerph-19-01333-t001:** Themes and sub-themes in a framework for recommendations.

Themes and Sub-Themes
Theme 1 A: Key recommendations for adolescents
Adolescents should do:
60 min of Physical Activity (PA) activity per day for 3–5 days.
Parents should participate for 150–300 min for 3–5 days per week.
20-min bouts of PA have a cumulative effect.
Theme 1 B: Key recommendations for parents
Help adolescents to set realistic PA goals.
Make use of a variety of PA options.
Free-play remains a popular option.
Parents to create opportunities for social PA settings.
Get involved and set realistic goals.
Theme 3: Resource for parents: Physical activity preferences
Type, examples, duration intensity, level of PA (Activity-Table included below)
Theme 4 A: Parental involvement Recommendations
Be actively involved in school sport.
Be involved with planning for PA and preparation before events.
Go and watch them participate (spectator parents).
Theme 4 B: Parental involvement Recommendations
Show an interest in what adolescents do.
Respect adolescent’s choices PA.
Participate in PA with your adolescent.
Allow adolescents to choose/select their activities.
Parents support adolescents with sports equipment, financial support, transport fees, support with overcoming barriers to participation.
Personal protective gear is something worn by a person to protect a specific body part: (helmets, eyewear, goggles, shin-guards, elbow and knee pads, and mouth-guards masks).
Theme 4 C: Parental involvement Recommendations (Directing behavior)
Set realistic boundaries for PA participation.
Open communication with adolescents is needed to establish: Ground rules, Curfews, Boundaries, PA time, Limit screen time/sedentary time.
Parents listen to your adolescent too they are savvy and knowledgeable.
Theme 4 D: Parental involvement Recommendations (Parental encouragement)
Provide positive feedback and motivate adolescents.
Good role models in parents, caregivers, and teachers should model and encourage an active lifestyle for children.
Praise, reward, and encourage adolescents to be active.
Being active as a family is a great way to model and encourage physical activity.
Theme 4 E: Parental involvement Recommendations (Parental awareness of the benefits)
Improved cognitive function.
Reduced risk of cancer.
Brain health benefits and improved cognitive function.
Reduced anxiety and depression risk.
Improved sleep and quality of life.
Both aerobic and muscle-strengthening physical activity is beneficial.
The health benefits for people with chronic and health conditions.
endorphins/feel good/self-concept/self-image
Theme 5 A: Safe PA in stressed environments
PA risks must be understood by parents.
Select types of PA that are appropriate for the level of fitness.
Screen the PA environments for safety risks. know what they want to do.
Consult a health care professional before starting with PA if adolescents have health conditions (types and amounts of PA).
Theme 5 B: Safe PA in stressed environments (Personal safety)
Parents assisting adolescents to sensible choices (when, where, how to be active).
Teach adolescents to be aware of their surroundings and to be alert and avoid risky situations.
Establish a buddy system for PA.
Join a walking bus to and from school.
Theme 5 C: Safe PA in stressed environments (Personal safety)
Physical separation from motor vehicles and awareness of surroundings.
Choose places that are well lit.
Following rules and safety rules is the best way to reduce activity-related injuries.
Theme 6 A: Sustained PA (Taking-Action)
Provide time for both structured and unstructured PA
PA through break time/recess.
The use of technology and digital tools to use during PA sessions is preferred by adolescents.
Online activities yoga, games, programs would interest adolescents
Theme 6 B: Sustained PA recommendations
Start gradually and be consistent.
Start PA at a young age and make it a lifestyle.
Everyone has a role to play: Schools and Communities Faith groups Businesses Civic organizations Parent-teacher associations Health groups and Public safety agencies Policymakers

**Table 2 ijerph-19-01333-t002:** Physical activity resource.

Type of Activity Including Definition and Example	Frequency (How Often)	Duration (How Long to Do the Activity)	Intensity (How Hard to Work)
Aerobic activities are defined: activities that make the heart beat faster. Example: run, brisk-walk, walk, swim, hiking, dance, free-play, skipping, gymnastics	3 days Per week	60 min per session, or, 20 min-bouts of activity repeated 3 times.	Moderate or vigorous activity depends on your fitness level. Start slow and build up gradually.
Muscle-strengthening activities are defined: activities linked to strengthening the muscles. Examples: climbing activities making use of one’s body weight.	2 days Per week	20 min-bouts of activity repeated 3 times. Start slow and build up gradually.	Moderate or vigorous activity depends on your fitness level. Start slow and build up gradually.
Bone strength activities are defined as activities selected by adolescents in the findings of the study and categorized here as bone strength. Examples: jumping jacks, running, brisk walking, activities using one’s body weight, pushing and lifting activities, moderate and vigorous housework, Tennis, hopscotch, and free-play.	2 days Per week	20 min-bouts of activity repeated 3 times. Start slow and build up gradually.	Moderate or vigorous activity
Balance and flexibility activities: activities preferred by adolescents in the current study and categorized here as balance and flexibility activities. Examples: movement and rhythmic movement activities, dance, gymnastics, whole-body stretching, walking the line, walking backwards in free-play, balancing on one leg, proprioception, balancing on foam.	Do these activities daily	bouts of activity, start slow and build up gradually.	Moderate or vigorous activity
Warm-up and cool-down are defined as light activities used to prepare the body for an activity session. It can take the form of active warm-up or passive warm-up activities. Example: walking on the sport, cycling slow and gradually warming the body up. The aim is that the heart rate must increase gradually. Cooling down helps to lower the heart rate. Cooling down and stretching combine to help alleviate muscle aches and pains.	Daily before and after every session	5–15 min before and after every session.	Light

## Data Availability

Data supporting reported results can be found, on the University of the Western Cape repository. including links to publicly archived datasets analyzed or generated during the study.

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
