# Peer review of "Recommendations to Enhance Parental Involvement and Adolescent Participation in Physical Activity"

_ijerph, 2022, doi:10.3390/ijerph19031333_

Round 1

Reviewer 1 Report

The study design has serious flaws, such as poorly designed study questions and improper methodology. The manuscript does not provide sufficient advance or impact for the journal.

General comment: The authors tried to develop some key recommendations for increasing adolescent participation in physical activity and parental involvement. They used a consensus workshop to develop the recommendations with stakeholder inputs including experts and parents working in the field of physical activity, physical education. However, the number of participants in the current study is not sufficient to reach a concrete conclusion. Also, in the sense of science, the scientific merit of the manuscript is very low.

Some of my specific comments are given below-

Specific comments:

Title

It's better to use “recommendations” rather than guidelines.

Abstract

In the abstract, the aim of the study clearly mentioned, methods also described. However, major results and findings are not properly presented.

Introduction

The introduction is one of the main sections of a research article. Unfortunately, the introduction section of the current manuscript is frustrating, not organized and does not provide sufficient background of the study. Authors should completely revise the introduction section. Please discuss the importance of physical activity for adolescent, problems associated with low or less physical activity, importance of parental involvement in adolescent physical activity, provide data or statistics on current scenarios/status of adolescent physical activity. It is very important to provide sufficient background information to readers so that they could understand your scientific contribution in the society. Authors should also explain the importance and necessity of their works.  Please check the citation in the text (i.e., line 31).

Methods and theoretical framework

The participant in the study is very small (only 20). Also, no adolescent participation.

Results

The results are not well explained. Please explain your results in various viewpoints more comprehensively but concisely.

Discussion

The discussion section could be more focused. The discussion doesn’t properly reflect the results obtained in this study. Authors can give more emphasis to discuss their findings from multiple angles in this section. Major limitations of the present study were not mentioned and opportunities to inform future research were not properly addressed.  

Conclusion

No major concern.

References

The authors are requested to check and review the references in the text and in the reference list at least four or more times. The reviewer thought that this section is enough for getting a rejection of the manuscript.

Overall comments: The manuscript is not well-written. The study design has serious flaws (sample size very low, no adolescent participation). The manuscript does not provide sufficient advance or impact for the journal.

Author Response

Dear reviewer kindly find attached a major review of the manuscript completed. I appreciate your input very much and your kind feedback. 

Reviewer 2 Report

TITLE

1. Is very good

ABSTRACT
is well written

MAIN TEXT

An article is well written

REFERENCES

Are generally fresh but need some corrections

MINOR COMMENTS

Accept after minor corrections

MINOR COMMENTS

In line 70 I recommend to add one sentence about preventing role of PA, among others, in Parkinson, Alzheimer, CD diseases including following papers

  1. PA and cardiovascular

http://www.aginganddisease.org/EN/10.14336/AD.2019.0516

  1. PA and AD

http://www.aginganddisease.org/EN/10.14336/AD.2019.0226

  1. PA and PD

In the intro it would be correct also to citate papars that indicate The effect of physical fitness and physical activity level on memory storage of Italian pre-adolescent secondary school students. Russo et al. You can find it here http://tss.awf.poznan.pl/files/2021/Vol%2028%20no%203/4_Russo_TSS_2021_283_195-202.pdf

Also it would be interesting to mention in the Intro that some Authors discuss WHO recommendations of PA as too conservative and too low

Gronek et al. Aging & Disease 2021

Details you will find here

http://www.aginganddisease.org/EN/10.14336/AD.2021.0107

Author Response

Dear reviewer thank you very much for the guidance provided with the manuscript. Kindly find attached major revisions done and completed in full with restructuring. 

Reviewer 3 Report

Attached document

Author Response

Dear reviewers 1-3, thank you very much for the feedback provided. I was able to restructure the article and I am able to resubmit the article after major revisions were done. Thank you very much that I could learn from the process

Round 2

Reviewer 1 Report

No additional comments. The authors tried their best to address all of my comments and concerns.